# Efficient Removal of Micropollutants by Novel Carbon Materials Using Nitrogen-Rich Bio-Based Metal-Organic Framework (MOFs) as Precursors

**Yazi Meng, Xiang Li * and Bo Wang**

School of Materials, Advanced Research Institute of Multidisciplinary Science, Beijing Institute of Technology, Beijing 100081, China
*   Correspondence: xiangli0369@bit.edu.cn

**Abstract:** Eliminating pharmaceuticals with trace concentrations in water is crucial in water purification. Developing an effective adsorbent for removing micropollutants from water has aroused great research interest. In this study, the feasibility of nitrogen-rich bio-based metal–organic framework (MOF)-derived carbon as an effective material to eliminate micropollutants from the water environment is discussed. A mixed ligand approach has been applied to synthesize IISERP-MOF27 successfully via the solvothermal method. Adenine, which is non-toxic, easily obtained, and cheap, was introduced into the structure. The novel heterogeneous porous carbon was produced by pyrolyzation with an extremely high surface area ($S_{BET}$ = 980.5 m$^2$/g), which is 12.8 times higher than that of pristine MOFs. Studies show that the highest surface area and abundant mesoporous structures ($V_{pore}$ = 0.496 cm$^3$/g) can be obtained when the MOFs are pyrolyzed at 900 °C. The saturated adsorption amount for sulfamethylthiazole (SMX) over MOF-derived carbon can reach 350.90 mg/g with a fast initial adsorption rate of 315.29 (mg/g·min). By adding the second linker adenine as the precursor, the adsorption performance for SMX was made extremely better than that of traditional active carbon (AC) and pyrolyzed ZIF-8(ZIF-8-C), one of the most classic Zn-MOFs. The adsorption capacity calculated by the Langmuir model ($R^2$ = 0.99) for SMX over bio-C-900 was 4.6 and 13.3 times more than those of AC and ZIF-8-C, respectively. The removal percentage of six representative pharmaceuticals can be well correlated to the structural parameter log $K_{ow}$ of each pharmaceutical, indicating the hydrophobic interaction should be one of the major mechanisms for the adsorption in water. This study offers a strategy to develop novel carbon materials to remove pharmaceuticals.

**Keywords:** pharmaceuticals; metal–organic framework; adsorption; performance

## 1. Introduction

In recent years, regular testing of micropollutants in tap water, river water, and municipal wastewater has received increasing attention [1]. This kind of pollutant includes antibiotics, anti-inflammatory medications, and a variety of chemicals [2]. They have adverse environmental impacts such as mutagenicity, carcinogenicity, aquatic toxicity, and other harmful effects on both humans and the ecosystem [3,4]. For instance, the frequently used antibiotic sulfamethoxazole (SMX) is difficult entirely eliminate in wastewater treatment plants (WWTPs), leading to the transformation of its metabolites and product residues into components of surface water [5]. It has been proven that prolonged exposure to such pollutants may have negative effects on individuals, such as liver failure and genetic damage [5,6]. Therefore, it is of great significance to remove PPCPs from aquatic ecosystems greenly and efficiently.

To remove these contaminants in water, several tertiary treatments have been used, such as photocatalytic degradation [7], the Fenton reaction [8], biodegradation [9], and membrane filtration [10]. Among these, adsorption technology, which is easy to operate and low-cost, has been considered a promising advanced water treatment process for eliminating

micropollutants in wastewater. To date, a large number of adsorbents have been investigated to remove micropollutants, including porous carbon [11], cyclodextrin polymer [12], and zeolites [13]. The carbon-based ones among these adsorbents are considered major materials in the water treatment process because of their high water/thermal stability and surface areas [14]. However, according to previous studies, their removal efficiency of SMX still needs to be improved compared to traditional materials (including activated carbon). Normally, about $54.34 \pm 2.35\%$ of SMX can be removed by treatment at WWTPs, but SMX content ranging from 0.3 ng $L^{-1}$ to 783 ng $L^{-1}$ is still detected after removal [15].

In recent years, MOFs with high specific surface areas and high porosities have been considered promising precursors to constructing adsorbents in water [16]. MOFs derived from carbon have been employed for the adsorption of water pollutants [17]. For example, results show that heavy metals, herbicides, and organic dyes can be effectively removed by carbon materials derived from Fe-MOFs [14,18–20]. Due to the especially low boiling point of the Zn atom, derived porous carbon can be created after the evaporation of metal based on Zn-based MOFs, such as ZIF-8, MOF-5, etc. [21–23]. On the other hand, bio-MOFs such as IISERP-MOF27(bio-27) and IISERP-MOF26 [24] with biological linkers should be promising in constructing MOFs because of their non-toxic, easily available, and inexpensive properties [25]. However, to the best of our knowledge, Zn-MOFs with mixed linkers-derived carbon have not been reported yet.

In this study, for the first time, bio-27-derived carbon was synthesized for the removal of micropollutants from water. The major characterization shows that this bio-MOF-derived carbon has obvious heterogeneous pores ranging from 1 nm to 6 nm. The precursors were pyrolyzed at 500~1000 °C (bio-C-500~bio-C-1000). The adsorption capacities and initial adsorption rates were evaluated by analyzing the removal performance for SMX over different bio-MOFs. The kinetic data were fitted using the Elovich model. Langmuir and Freundlich's models were applied to compare the capacities. The operational parameters were studied. To further evaluate the adsorption performance for pharmaceuticals with diverse structures, the adsorption behaviors for six pharmaceuticals—ketoprofen (KP), antipyrine (AT), ibuprofen (IBU), chloramphenicol (CAP), paracetamol (PC), and sulfamethoxazole (SMX)—were studied.

## 2. Experimental Section

### 2.1. Chemical Agents

$Zn(NO_3)_2 \cdot 6H_2O$ (CAS, 10196-18-6), dimethylformamide (DMF, 99%), and methanol (MeOH, CAS, 67-56-1) were purchased from Sinopharm (Beijing, China). Adenine (CAS, 73-24-5) and terephthalic acid (CAS, 100-21-0) were purchased from Energy Chemical (Beijing, China) (>99%). Active carbon was purchased from Energy Chemical Co. Ltd. (CAS, 7440-44-0). Sulfamethoxazole and other chemicals were all obtained from Sigma–Aldrich with a purity of >99%. Water was obtained from a Milli-Q system with 18.2 M$\Omega \cdot$cm$^{-1}$. The target compounds can be found in Table 1. $K_{ow}$, which is the ratio of the concentration of molecules in the octanol phase to their concentration in the aqueous phase, is widely used to describe the hydrophobicity of drugs [26]. A drug's greater hydrophobicity is indicated by a larger log $K_{ow}$ value [27].

**Table 1.** Pharmaceuticals used in this study.

| Full Name | Abbra. | Molecular Weight | Log $K_{ow}$ | Chemical Structures |
|---|---|---|---|---|
| ketoprofen | KP | 257.3 | 3.12 |  |
| antipyrine | AT | 189.1 | −1.55 |  |

**Table 1.** *Cont.*

| Full Name | Abbra. | Molecular Weight | Log $K_{ow}$ | Chemical Structures |
|---|---|---|---|---|
| ibuprofen | IBU | 205.1 | 0.45 | |
| chloramphenicol | CAP | 321.0 | 1.1 | |
| paracetamol | PC | 180.1 | 1.58 | |
| sulfamethoxazole | SMX | 254.1 | 0.89 | |

### 2.2. Synthesis of Bio-27 and Its Derivatives Bio-C Materials

Bio-27 and its derivatives were synthesized via a solvothermal method [24]. $Zn(NO_3)_2 \cdot 6H_2O$ (260.86 mg, 0.87 mmol), adenine (AD; 117.38 mg, 0.87 mmol), and terephthalic acid ($H_2BDC$; 72.60 mg, 0.435 mmol) were mixed with 10.8 mL DMF, $H_2O$, and MeOH (v/v/v = 1:1:0.6) and dispersed by ultrasonic processing for 30 min. Then, a white dispersion was obtained. The mixed solution was transferred to a 50 mL Teflon-lined autoclave and kept in an oven at 120 °C for 48 h. DMF and methanol were used to wash the sample before it was dried at 60 °C. A specific amount of bio-27 was evenly distributed in the center of the quartz boat and heated by 5 °C min$^{-1}$ for 3 h in nitrogen after 6 h of vacuum activation at 120 °C. The materials obtained at the different temperatures of T (T = 500~1000 °C) are named bio-C-T. As the furnace chamber cooled to room temperature, bio-C-T compounds were generated. Some basic characterization was conducted to select a suitable material for further application in removing pharmaceuticals from water. The schematic diagram can be found in Figure 1.

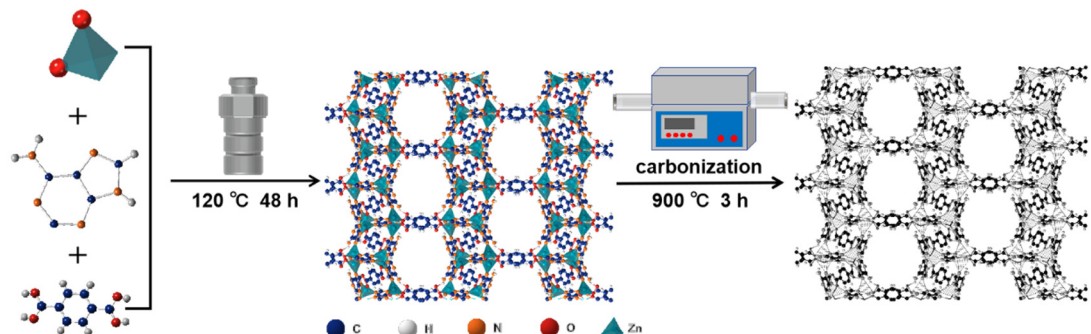

**Figure 1.** Schematic diagram of the synthesis of bio−27 and its derivatives, bio−C materials.

### 2.3. Synthesis of ZIF-8 and Its Derivatized Carbon-Based Materials (ZIF-8-C)

ZIF-8 was synthesized based on the previous report [28]. $Zn(NO_3)_2 \cdot 6H_2O$ (743.8 mg, 2.5 mmol) and dimethylimidazole (821.0 mg,10 mmol) were mixed with 50 mL MeOH and put into a 100 mL Shuniu bottle, where they were and continuously and evenly stirred for 24 h at room temperature. After standing for precipitation, the materials were separated and washed three times with fresh MeOH. Then, the materials were dried at 60 °C.

### 2.4. Characterization

The X-ray diffraction pattern was studied by a multipurpose, high-efficiency X-ray diffractometer (PXRD, Rigaku MiniFlEX600) ($\lambda$ = 0.154 nm). The scan rate was 10°/min at room temperature. The adsorption–desorption curve was measured by an automatic specific surface and pore analyzer (Quanta chrome ASiQMVH002-5). Materials were activated in vacuum at 120 °C for 12 h. The Brunel–Emmett–Teller (BET) equation was used for measuring the specific surface area. The chemical state was obtained by X-ray photoelectron spectroscopy (XPS, PHI QUANTERA-II SXM, ULVAC-PHI, USA), and the X-ray source was Al-Ka (1486.6 eV, line width 0.68 eV). The zeta potential of bio-C was tested using a nanoparticle size and zeta potential analyzer (DLS, Malvern Zetasizer Nano ZS90, Worcestershire, UK).

### 2.5. Adsorption Experiment

The adsorption experiments for removing pharmaceuticals with two concentrations, 10 mg/L and 400 µg/L, were conducted separately. The solution was prepared with ultrapure water. Amounts equal to 4.5 mg of different adsorbents were added to a 50 mL glass vial containing 50 mL of a simulated solution of PPCPs at pH 7.0. All the batch adsorption experiments were performed on a multi-point stirrer at 500 rpm and 25 °C. At each time interval, the solution was sampled and then filtered with a PES filter. The adsorption experiments were conducted for 45 min. The pseudo-second-order kinetic model is shown in Equations (1) and (2), where $Q_t$ represents the adsorption amount at time t, and µg/g is the unit. $Q_e$ is the equilibrium adsorption amount, and µg/g is the unit. $v_0$ represents the initial adsorption rate, and µg/(g·min) is the unit. k is the intraparticle diffusion rate constant.

$$\frac{t}{Q_t} = \frac{1}{v_0} + \frac{t}{Qe} \tag{1}$$

$$v_0 = k \times Q_t^2 \tag{2}$$

### 2.6. Instrumental Analysis for Pharmaceuticals

Pharmaceuticals with concentrations of 10 mg/L were separated and detected using HPLC-UV with an Agilent 1260 system. The column was an Agilent SB-C18 column (2.7 µm, 4.6 mm × 150 mm). The mobile phase was water containing 0.1% formic acid and 1 mM ammonium acetate (($NH_4$)$_2$AC) (60%), as well as MeOH (40%). The temperature in the column was 30 °C. The UV light's detecting wavelength was 270 nm. Triple quadrupole mass spectrometry (Agilent 1290, 6465 QQQ, Santa Clara, CA, USA) and ultra-performance liquid chromatography were used to measure the drugs with trace concentrations. The column was an Eclipse Plua C18 RRHD (2.1 × 50 mm 1.8 µm). The mobile phases were water with 0.1% formic acid and 2 mM ammonium acetate (A) and acetonitrile (B). The flow rate was set at 0.2 mL/min. The dilute gradient was 0–2 min (10% B), 2–10 min (10~45% B), 10~13 min (45~90% B), 13~14 min (90% B), and 14.1 min (10% B). The injection volume was 20 µL. The flow rate was 0.2 mL/min. The peak areas of each pharmaceutical were quantitively analyzed by the Quant-My-Way software, developed by Agilent.

## 3. Results and Discussion

### 3.1. Characterizations of Bio-C-T

3.1.1. Structural and Porous Properties

As shown in Figure 2a, the PXRD patterns of bio-27 were consistent with the simulation, with the main characteristic peaks at 5°, 9°, and 15° [24], indicating the metal–organic framework (MOF) called bio-27 was synthesized successfully. The PXRD pattern of bio-C-T can be found in Figure 2b, indicating that all the synthesized carbons are in an amorphous state. It is to be noted that bio-C-800, 900, and 1000 showed two broad peaks at 24° and 43°, corresponding to the diffraction peaks of the (002) crystal plane and (100) crystal plane, respectively [29]. The observed peaks move slightly to smaller angles with increased activation temperatures, indicating the distance of the interlayer could gradually increase.

According to previous studies, this shift could be caused by defects during carbonization [30]. However, bio-C-500 may not be carbonized completely based on the positions of characteristic peaks in the PXRD spectrum. On the other hand, the colors of bio-C-800, bio-C-900, and bio-C-1000 are black, consistent with the color of carbon black. This could also confirm the completeness of the carbonization of the four materials. The nitrogen adsorption isotherm in Figure 2c reveals a mixture of type I and IV isotherms. Obviously, the adsorption curve of bio-C-900 has a hysteresis loop, indicating that the material has a hierarchical porous structure and a relatively more mesoporous structure.

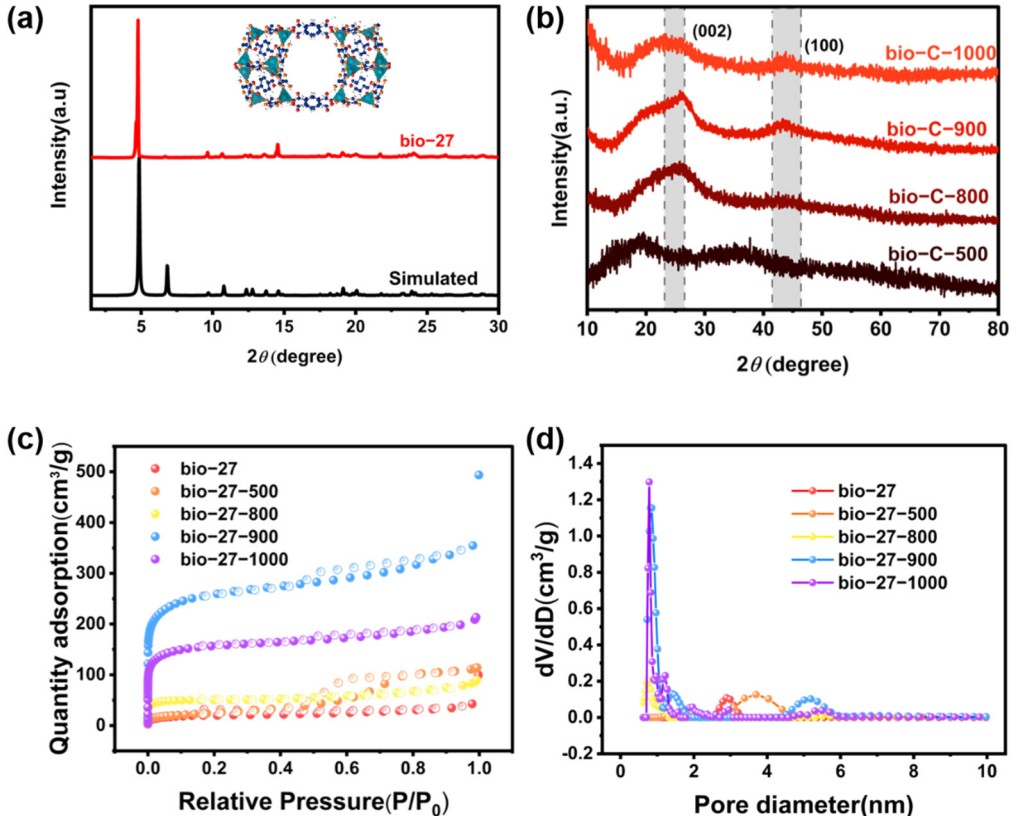

**Figure 2.** (**a**) PXRD patterns of IISERP−MOF−27 and its (**b**) derived carbon materials, which are pyrolyzed at temperatures ranging from 500 °C to 1000 °C. (**c**) The nitrogen adsorption and desorption isotherms of bio−27 and bio−C−T. (**d**) The pore size distribution of the as-prepared samples of bio−27 and bio−C−T (T = 500~1000 °C).

In comparison, the hysteresis loops of bio-C-800 and bio-C-1000 are not obvious, and the adsorption curves are more in line with the type I adsorption isotherm. The surface area and pore volume show an upward trend with increased carbonization temperature. Still, the specific surface area of bio-C-1000 was lower than that of bio-C-900, which may be due to the partial collapse of the mesoporous structure at high temperature due to the complete evaporation of Zn atoms (Table 2).

**Table 2.** Calculated surface area ($S_{BET}$) and porous structures.

| Materials | $S_{BET}$ (m$^2$/g) | $V_{total}$ (cm$^3$/g) | Pore Width (nm) |
|---|---|---|---|
| Bio-27 | 76.5 | 0.054 | 2.890 |
| Bio-27-500 | 90.2 | 0.160 | 3.698 |
| Bio-27-800 | 204.2 | 0.108 | 0.783 |
| Bio-27-900 | 980.4 | 0.496 | 0.852 |
| Bio-27-1000 | 605.3 | 0.294 | 0.783 |

The pore size distribution curve calculated in the microporous region (Figure 2d) shows that all catalysts are dominated by micropores with a pore size of 0.8 nm. Although the microporous area of bio-C-500 accounts for a small proportion, the pore volume of bio-C-500 is higher than that of bio-C-800 due to the wider distribution of the mesoporous area. The surface area of the pure MOF (bio-27) is 76.4 $m^2$/g, as shown in Table 2. However, bio-27-C has much higher surface areas ranging from 204.2 $m^2$/g to 980.4 $m^2$/g.

On the other hand, the pore volume of bio-27-C-900 is the greatest, with a value of 0.496 $cm^3$/g. The pore width of the bio-C-900 is the greatest, with a value of 0.852 nm, among all the materials. The well-developed pore structure in the adsorbent may provide more adsorption sites for SMX, which should be beneficial for reducing diffusion resistance.

### 3.1.2. Morphology of the Samples

In Figure 3a,b, the precursor bio-27 has an irregular two-dimensional sheet structure, but the thickness is too large. With increasing pyrolysis temperatures, the edge morphology of bio-C-500 becomes irregular (Figure 3c), but it can be observed that it exists in the form of sheet-like stacking, and the scale becomes thinner. While bio-C-800, bio-C-900, and bio-C-1000 (Figure 3d–f) exhibit changes in longitudinal scale, the edges are more regular, and the ordered sheet-like stacking structure is favorable for the mass transfer process [31]. The jagged edges of bio-C-1000 may be due to the collapse of the internal structure. HRTEM was used to further characterize the morphology and microstructure of the catalyst. In the bright field image (Figure 3g), it can be observed that the surface of the catalyst with sheet-like morphology has uniformly distributed mesopores and only short-range ordered ones. N-doped carbon fringes do not exhibit any long-range ordered lattice fringes, indicating no crystalline species in the catalyst, which is mutually confirmed by the results of PXRD, as discussed above. The results show that the prepared bio-C-900 material has abundant pore structures.

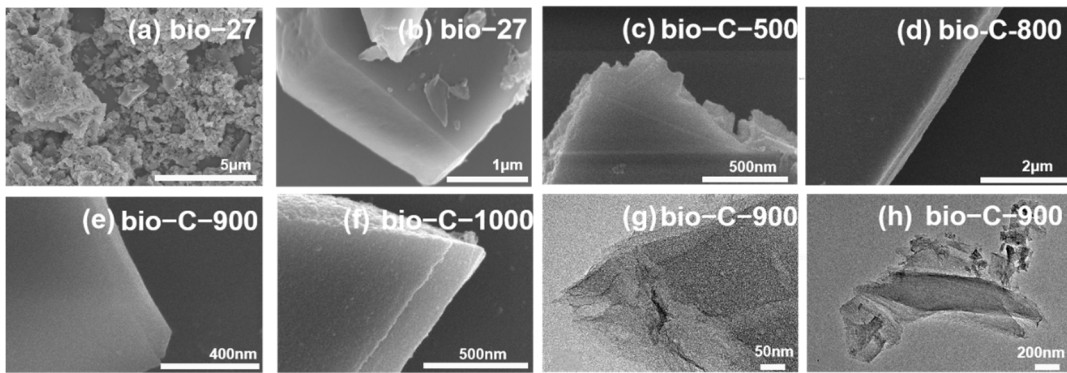

**Figure 3.** (**a**–**f**) SEM images of bio−27 and bio−C−T. (**g**,**h**) HRTEM of bio−C−900.

### 3.1.3. Fourier Transform Infrared (FTIR) Spectrum of Bio-27-Derived Carbon

Analysis via FTIR spectroscopy was carried out in order to investigate the various surface functional groups. According to Figure 4, the stretching vibrations of O-H and N-H can be found at around 3400 $cm^{-1}$ [32]. The band at 2100 $cm^{-1}$ is ascribed to the C=C=C stretching vibration [33]. The C=O stretching vibration is responsible for the band at 1650 $cm^{-1}$ [32], while the N-H in-plane and out-of-plane bending vibrations are responsible for the bands at 1580 $cm^{-1}$ and 850 $cm^{-1}$, respectively [32]. Furthermore, the bands at 1380 $cm^{-1}$ and 1070 $cm^{-1}$ are attributed to C-O stretching vibrations and C-N stretching vibrations [32,34], respectively, indicating the existence of nitrogen and oxygen functional groups on bio-C.

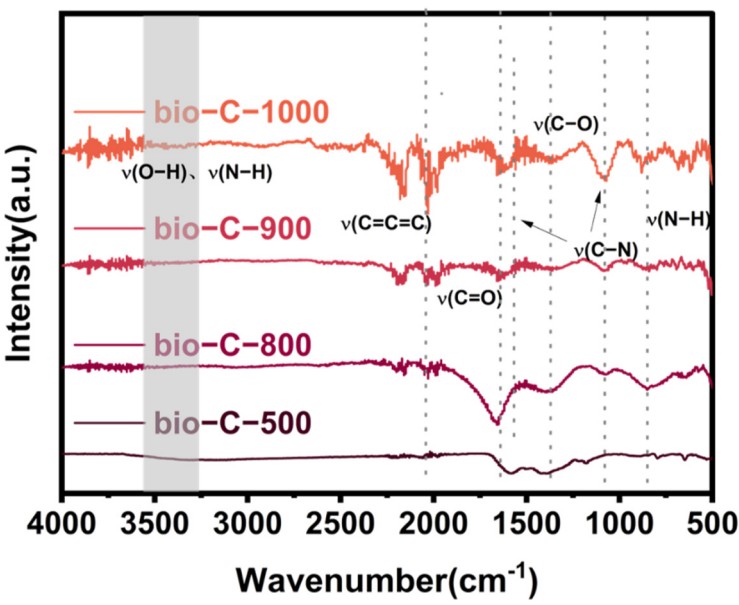

**Figure 4.** Fourier transform infrared (FTIR) spectrum of IISERP−MOF−27−derived carbon which is pyrolyzed at temperatures ranging from 500 °C to 1000 °C.

### 3.2. Adsorption Experiments

3.2.1. Pseudo-Second-Order Adsorption Kinetics of SMX

Within a contact time of 45 min, the impact of bio-C-T materials at various pyrolysis temperatures on SMX adsorption was examined. The results are displayed in Figure 5a. To fit the experimental data, a pseudo-second-order kinetic model was applied, and the relevant fit plots were produced. The model compound's adsorption of the adsorbent is indicated by the variables $Q_t$ and $Q_e$ in the equation at reaction time t (min)—when the reaction achieves saturation equilibrium. $Q_e$ and $v_0$ represent the adsorption amount and the initial adsorption rate (mg/g·min), respectively. According to Table 3, bio-C-900 has the highest $Q_e$ (100.85 mg/g) and the best adsorption capacity of SMX among the series of bio-C-T materials. In addition, its $v_0$ (315.29 mg/g·min) is the highest among all the prepared materials. The mass transfer resistance of bio-C-900 to SMX is substantially lower than that of the manufactured bio-C at other temperatures, and its adsorption sites are more readily accessible to SMX. Additionally, its $v_0$ is more than 30 times that of bio-C-1000. We therefore hypothesize that surface area has a close relationship with the adsorption amount of SMX in this study. A higher surface area should favor a higher adsorption amount.

**Table 3.** Pseudo-second-order adsorption kinetics fitting values.

| Materials | k (g/mg·min) | $Q_e$ (mg/g) | $v_0$ (mg/g·min) | $R^2$ |
|---|---|---|---|---|
| Bio-27-500 | —— | 1.65 | —— | 0.99 |
| Bio-27-800 | 0.0059 | 27.27 | 4.38 | 0.98 |
| Bio-27-900 | 0.031 | 100.85 | 315.29 | 0.93 |
| Bio-27-1000 | 0.0021 | 70.79 | 10.52 | 0.98 |

On the other hand, large pore sizes could enhance the adsorption kinetics [35]. Similar results can be found in the adsorption process by other porous material [36]. For example, Guo found that the introduction of mesopores above 2 nm can greatly increase the pore volume. The graphene oxide/carbon composite nanofibers with abundant mesopores prepared in this study achieved improved adsorption capacity for volatile organic compounds (VOCs), and the highest adsorption capacity values for benzene and butanone reached 83.2 cm$^3$ g$^{-1}$ and 130.5 cm$^3$ g$^{-1}$, respectively. According to Table 2, we can know that the

pore volume of bio-C-T-series materials was also greatly improved after the appearance of mesopores, which is consistent with the experimental results.

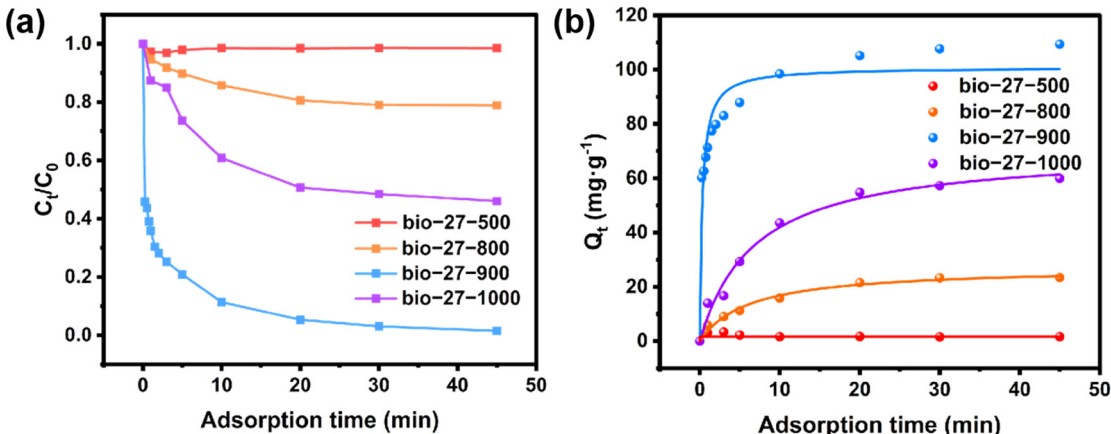

**Figure 5.** (**a**) Adsorption kinetics curves of bio−C−T. (**b**) The corresponding fitted pseudo-second-order adsorption kinetics. Reaction conditions: SMX = 10 mg/L, reaction volume = 50 mL, catalyst dosage = 180 mg/L, initial solution pH = 7, temperature = 25 °C, reaction time = 45 min).

### 3.2.2. Effect of Reaction Time

The elimination of SMX was compared over bio-C-900, AC, and ZIF-8 (Zn as the metal center). The ZIF-8-C obtained after carbonization at the same temperature was applied in an adsorption experiment for removing SMX with an initial concentration of 10 ppm for 60 min. As shown in Figure 6a, bio-C-900 showed ultra-fast adsorption behavior, with 55% removal within only 15 s. A higher removal rate of 90% could be achieved within only 10 min. After reaching adsorption saturation, the removal rate of SMX by bio-C-900 can reach 98.7%. In comparison, the removal rates for SMX over AC and ZIF-8-C were 36.5% and 9.7%, respectively. Two models, the Elovich model and the second-order kinetic model, were applied in Table 4. The Elovich kinetic fitting formula (Equation (3)) is as follows:

$$Q_t = A + b \times \ln t \tag{3}$$

where A and b are both Elovich constants. The fitting degree of the two models is very high, indicating that the adsorption process of the adsorbent includes both chemisorption with electron transfer and electron transfer during the pseudo-second-order kinetic process.

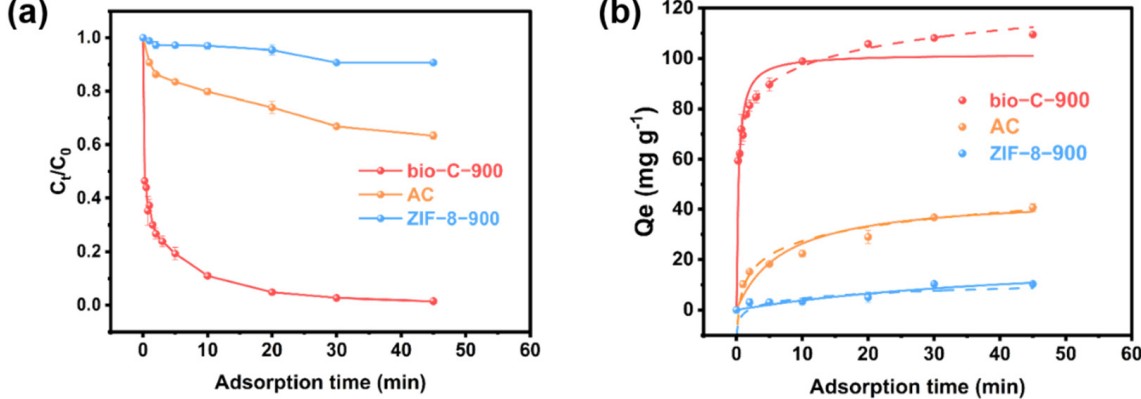

**Figure 6.** (**a**) Adsorption kinetics curves of all materials for SMX in this study; (**b**) The corresponding pseudo-second-order adsorption kinetic model and Elovich kinetic model were fitted. Reaction conditions: initial concentration of SMX = 10 mg/L, reaction volume = 50 mL, catalyst dosage = 4.5 mg, initial solution pH = 7, temperature = 25 °C, reaction time = 45 min.

**Table 4.** Langmuir and Freundlich adsorption isotherm model.

| | Langmuir Model | | | Freundlich Model | | |
|---|---|---|---|---|---|---|
| **Sample** | **$Q_{max}$ (mg/g)** | **$K_L$ (L/mg)** | **$R^2$** | **$K_F$ (mg g$^{-1}$(L mg$^{-1}$)$^{1/n}$)** | **n** | **$R^2$** |
| Bio-C-900 | 350.90 | 0.061 | 0.99 | 53.23 | 2.47 | 0.96 |
| AC | 76.96 | 0.025 | 0.98 | 29.79 | 4.03 | 0.82 |
| ZIF-8-C | 26.44 | 0.027 | 0.99 | 10.88 | 4.32 | 0.88 |

### 3.2.3. Effect of Initial Concentration

The amount of adsorption increases with increasing initial concentration and reaches saturation for bio-C-900, AC, and ZIF-8-C, as well as for adsorbents over the studied concentration (Figure 7). At initial concentration of 20 mg/L, 86.6% of SMX could be removed by the bio-C-900 sorbent. When the initial concentrations were 40, 50, 60, and 80 mg/L, SMX was partially removed with efficiencies of 58.3%, 53.1%, 44.8%, and 32.7%, respectively. Such a phenomenon can be explained by the limited adsorption sites of the same adsorbent. The adsorption capacity increases with a higher concentration of adsorbate. This demonstrates that the initial concentration should limit the adsorption amount because the adsorption sites are sufficient [37]. In addition, the data were fitted using the Langmuir and Freundlich equations.

$$Q_e = Q_{max} \times \frac{K_L C_e}{1 + K_L C_e} \tag{4}$$

$$Q_e = K_F C_e^{\frac{1}{n}} \tag{5}$$

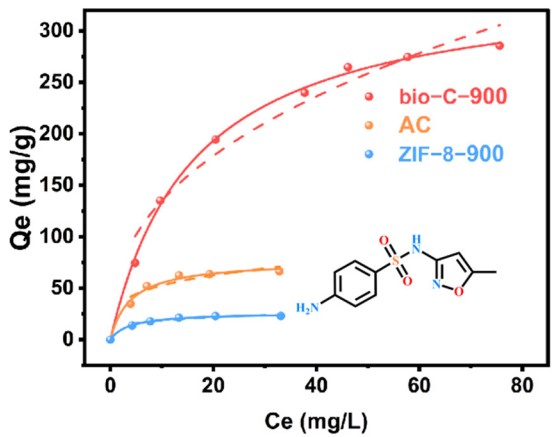

**Figure 7.** Adsorption isotherm for the removal of SMX over the three materials: active carbon (AC), ZIF−8−C, and bio−C−900. Reaction conditions: initial concentration of SMX = 5~80 mg/L, initial solution pH = 7, temperature = 25 °C, reaction time = 600 min.

The adsorption capacity of SMX over bio-C-900 was 4.6 and 13.3 times higher than those of AC and ZIF-8-C, respectively. The $K_L$ value indicates that SMX is more easily removed by bio-C-900, confirming that the prepared bio-C-900 could have relatively better adsorption properties than those of conventional materials. The adsorption of SMX over bio-C-900 tends to be homogeneous monolayer adsorption [38] because the Langmuir model linear fitting result ($R^2$ = 0.99) is slightly better than that in the Freundlich model ($R^2$ = 0.96). Therefore, the material tends towards homogeneous monolayer adsorption. In addition, the maximum adsorption capacity of SMX over bio-C-900 (350.90 mg/g) was 3.6 times higher than that over AC (76.96 mg/g), which further proved that bio-C-900 is conducive to the transport of substrates on its surface and pores. The heterogeneous multi-layer adsorption effectively improves the adsorption capacity and adsorption kinetics. The specific fitting data are shown in Table 4 below.

### 3.3. Exploring the Adsorption Mechanism for Micropollutants

To gain more insight into the reaction mechanism, the effect of solution pH (3~9) was evaluated in Figure 8a. As seen in Figure 8, the removal percentages of SMX varied significantly with initial pH values. SMX is an amphiphilic molecule with two $pK_a$ values ($pK_{a1}$ = 1.8 and $pK_{a2}$ = 5.6) [39]. When the solution pH is higher, SMX is prone to exist in its deprotonated form with a negative charge. The zeta potential can be found in Figure 8b. As a result, the adsorption amount decreased obviously, probably because of electrostatic expulsion. However, in an acid environment, SMX is usually protonated, with a positive charge, which could lead to a decrease in electrostatic interaction. However, the adsorption performance for the removal of SMX is higher at pH values ranging from 3 to 5. Thus, several major mechanisms could co-exist to contribute to the high adsorption amount. For example, the solubility of SMX is lower in an acid environment, and it is very slightly soluble in water but is soluble in alkali hydroxides, as previously reported [40,41]. The properties of hydrophilic interaction should be considered.

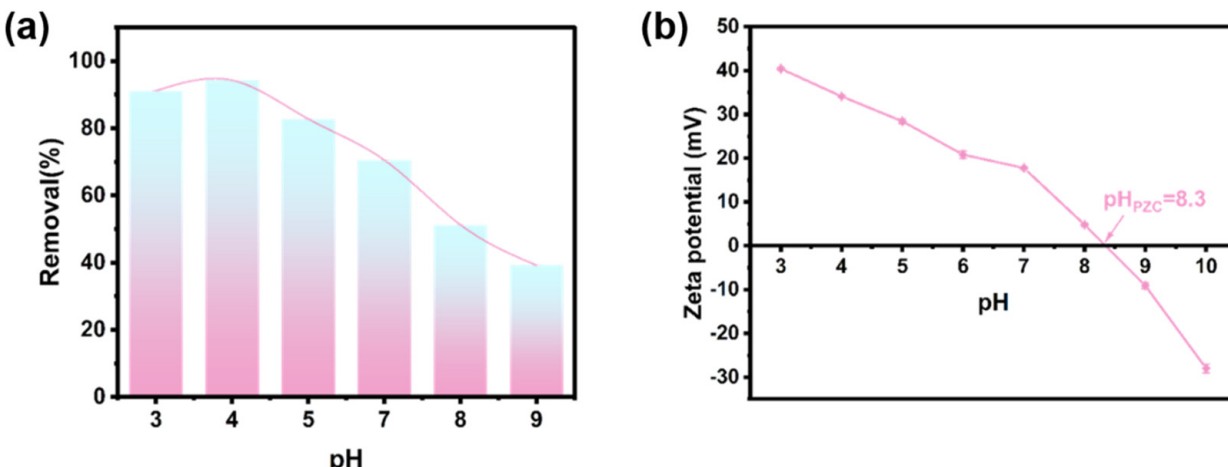

**Figure 8.** (**a**) Effect of pH on SMX removal. (**b**) Zeta-potential of bio−C−900.

To further evaluate the adsorption performance for pharmaceuticals with diverse structures, the adsorption behavior for six pharmaceuticals—ketoprofen (KP), antipyrine (AT), ibuprofen (IBU), chloramphenicol (CAP), paracetamol (PC), and sulfamethoxazole (SMX)—were studied. Obviously, we found that the adsorption performance (removal percentages within 45 min reaction time) correlated well ($R^2$ = 0.97) with the physical–chemical parameter log $K_{ow}$ of each pharmaceutical. The log $K_{ow}$ values of ketoprofen (KP), antipyrine (AT), ibuprofen (IBU), chloramphenicol (CAP), paracetamol (PC), and sulfamethoxazole (SMX) were 3.12, −1.55, 0.45, 1.1, 1.58, and 0.89 [42–45], respectively. Tung Xuan Bui et al. found a linear relationship between log Kow and adsorption amount at a wide range of pH values for 12 drugs, indicating that the adsorption of drugs on TMS-SBA-15 is mainly driven by hydrophobic interactions [46]. Taku Matsushita [47] et al. evaluated the adsorption capacity of nine activated carbons for geosmin and 2-methylisoborneol (MIB) and showed that hydrophobic materials enhance the adsorption. Based on the above discussion, hydrophobic interactions are dominant in this adsorption mechanism. Based on the discussion above, the proposed mechanism is shown in Figure 9b.

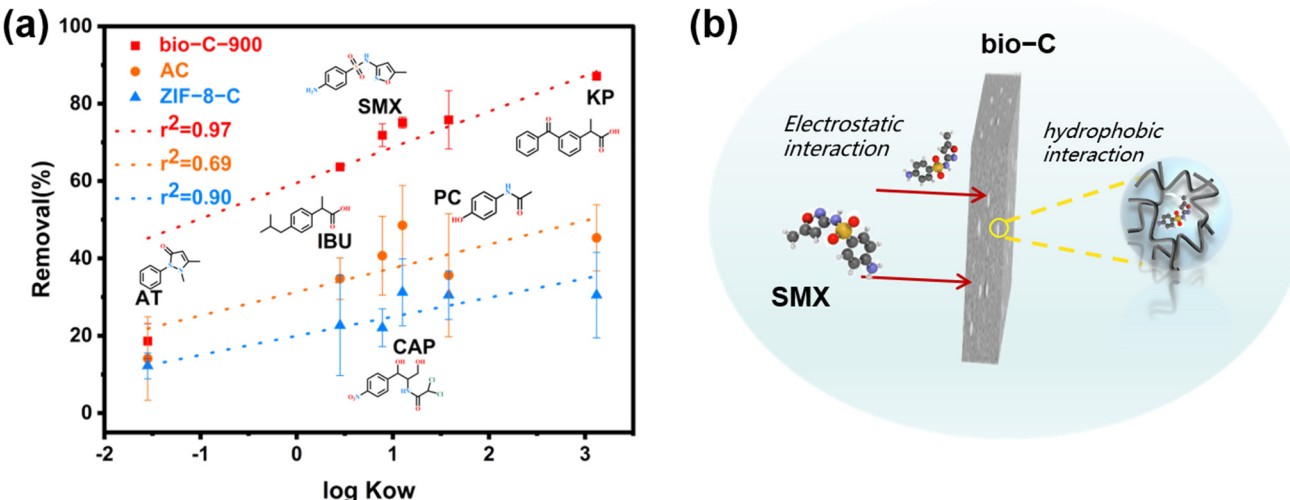

**Figure 9.** (**a**) Effect of $K_{ow}$ on diverse micropollutants. (**b**) Proposed reaction mechanism.

### 4. Conclusions

In summary, a novel MOF-derived carbon has been designed and synthesized for the first time. A mixed ligand approach has been applied to successfully synthesize bio-27 via the solvothermal method. By pyrolyzation at different temperatures, the optimal condition at 900 °C was selected to generate a novel heterogeneous porous carbon with the highest surface area ($S_{BET}$ = 980.5 m$^2$/g) and a large pore volume (0.496 cm$^3$/g). The maximum saturated adsorption capacity for sulfamethylthiazole (SMX) over MOF-derived carbon can reach 351.3 mg/g with a fast initial adsorption rate of 315.29 (mg/g·min). The adsorption capacity calculated by the Langmuir model ($R^2$ = 0.99) for SMX over bio-C-900 was 4.6 and 13.3 times more than the values for AC and ZIF-8-C, respectively. Hydrophobic interaction should be one of the major mechanisms for the material's adsorption in water. This study offers a strategy to develop novel carbon materials to remove pharmaceuticals from water.

**Author Contributions:** Conceptualization, X.L. and Y.M.; methodology, X.L. and Y.M.; software, Y.M.; validation, X.L. and Y.M.; formal analysis, X.L. and Y.M.; investigation, X.L. and Y.M.; resources, B.W.; data curation, X.L. and Y.M. writing—original draft preparation, Y.M.; writing—review and editing, X.L; visualization, X.L; supervision, B.W.; project administration, X.L. and B.W.; funding acquisition, X.L. and B.W.; All authors have read and agreed to the published version of the manuscript.

**Funding:** This work was financially supported by the National Natural Science Foundation of China (Grants 21625102, 21971017, and 21906007), China's National Key Research and Development Program (Grant 2020YFB1506300), and the Beijing Institute of Technology Research Fund Program. We gratefully acknowledge the Analysis and Testing Center of the Beijing Institute of Technology.

**Conflicts of Interest:** The authors declare no conflict of interest.

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
