# Peer review of "Efficient Removal of Micropollutants by Novel Carbon Materials Using Nitrogen-Rich Bio-Based Metal-Organic Framework (MOFs) as Precursors"

_water, doi:10.3390/w14213413_

Round 1

Reviewer 1 Report

The article entitled: “Efficient removal of micropollutants by novel carbon materials using a nitrogen-rich bio-based metal-organic framework (MOFs) as precursors” is based on the comparison of new materials to improve the removal of micropollutants from water. The title is clear, language is correct, and the topic is very attractive. Nevertheless, it needs some minor improvements before it can be published in Water.

In detail, I have the following comments.

The abstract and introduction are clear, they present the subject and the interest of this work.

The material and methods part is well written and interesting, but the composition of solvent B for the HPLC-MS experiment is missing.

In the result and discussion section, there are errors in the names of the figures in the text (e.g., line 172, Table 1 should be Table 2) please check all figure and table numbers. The proposed mechanism in figure 9 could be more detailed in the text.

The conclusion is clear, but could be a little less concise.

In conclusion, the topic is very interesting and promising, for all the reasons cited above, this article should include a minor revision before publication in Water.

Author Response

Response to Reviewer 1 Comments

Comment 1:

The article entitled: “Efficient removal of micropollutants by novel carbon materials using a nitrogen-rich bio-based metal-organic framework (MOFs) as precursors” is based on the comparison of new materials to improve the removal of micropollutants from water. The title is clear, language is correct, and the topic is very attractive. Nevertheless, it needs some minor improvements before it can be published in Water.

In detail, I have the following comments.

Response 1:

We appreciate the summary of Reviewer 1. Please refer to the following part of this letter: a point-by-point response to specific comments from Reviewer 1, highlighted in yellow in the revised manuscript.

Comment 2:

The abstract and introduction are clear, they present the subject and the interest of this work.

The material and methods part is well written and interesting, but the composition of solvent B for the HPLC-MS experiment is missing.

Response 2:

We appreciate the kind suggestions of reviewer 1. The solvent B is acetonitrile and we have revised the gradient elution protocol.

Page 7, lines 169-172 in the revised manuscript now reads:

The mobile phases were water with 0.1% formic acid and 2 mM ammonium acetate (A) and acetonitrile (B). The flow rate was set at 0.2 mL/min. The dilute gradient was 0 min – 2min (10% B), 2 min -10 min (10 % ~ 45 % B), 10 min~13 min (45%~90% B) 13 min ~14 min (90 % B), 14.1 min 10% B.

Comment 3:

In the result and discussion section, there are errors in the names of the figures in the text (e.g., line 172, Table 1 should be Table 2) please check all figure and table numbers. The proposed mechanism in figure 9 could be more detailed in the text.

Response 3:

We appreciate the kind suggestions of reviewer 1. We have fixed the error here and explained the mechanism in more detail.  With these two studies (J. Hazard. Mater. 254-255 (2013) 345-353. Water Res. 85 (2015) 95-102.), we illustrate the respective roles assumed by the hydrophobicity of the drug and the material between the hydrophobic effect and the amount of adsorption. We have added the discussions in the revised manuscript.   

Page 21-22, lines 525-531 in the revised manuscript now reads:

[46] T.X. Bui, V.H. Pham, S.T. Le, H. Choi, Adsorption of pharmaceuticals onto trimethylsilylated mesoporous SBA-15, J. Hazard. Mater. 254-255 (2013) 345-353. https://doi.org/https://doi.org/10.1016/j.jhazmat.2013.04.003.

[47] Y. Matsui, S. Nakao, A. Sakamoto, T. Taniguchi, L. Pan, T. Matsushita, N. Shirasaki, Adsorption capacities of activated carbons for geosmin and 2-methylisoborneol vary with activated carbon particle size: Effects of adsorbent and adsorbate characteristics, Water Res. 85 (2015) 95-102. https://doi.org/https://doi.org/10.1016/j.watres.2015.08.017.

Page 8, lines 196-199 in the revised manuscript now reads:

Still, the specific surface area of bio-C-1000 was lower than that of bio-C-900, which may be due to the partial collapse of the mesoporous structure at high temperature due to the complete evaporation of Zn atoms the partial collapse of the mesoporous structure at a high temperature (Table 2).

Page 15-16, lines 350-355 in the revised manuscript now reads:

Tung XuanBui et al. performed a linear relationship between log Kow and adsorption amount at a wide range of pH for 12 drugs, indicating that the adsorption of drugs on TMS-SBA-15 is mainly driven by hydrophobic interactions [46]. Taku Matsushita [47] et al. evaluated the adsorption capacity of nine activated carbons for geosmin and 2-methylisoborneol (MIB) and showed that hydrophobic materials enhance the adsorption. Based on the above discussion, hydrophobic interactions are dominant in this adsorption mechanism.

Comment 4:

The conclusion is clear, but could be a little less concise.

Response 4:

We appreciate the kind suggestions of reviewer 1. The conclusion has been reivised.

Page 16, lines 368-371 in the revised manuscript now reads:

In summary, a novel MOFs derived carbon has been designed and synthesized for the first time. A mixed ligand approach has been applied to successfully synthesize bio-27 via the solvothermal method. By pyrolyzation at different temperatures, the optima condition at 900 °C was selected to generate a novel heterogeneous porous carbon with the highest surface area (SBET = 980.5 m2/g) and large pore volume (0.496 cm3/g). The maximum saturated adsorption capacity for sulfamethylthiazole (SMX) over MOF-derived carbon can reach 351.3 mg/g with a fast-initial adsorption rate of 315.29 (mg/g • min). The adsorption capacity calculated by the Langmuir model (R2 = 0.99) for SMX over bio-C-900 was 4.6 and 13.3 times more than AC and ZIF-8-C, respectively. Hydrophobic interaction should be one of the major mechanisms for the adsorption in water. This study offers a strategy to develop novel carbon materials to remove pharmaceuticals.

Reviewer 2 Report

This article tries to develop an effective adsorbent for removing micropollutants in water.
The experiments seem carefully performed and well explained.
However, the following points should be
clarified.  

1.     The expression of Eq. (5) for Freundlich equation is incorrect.

2.     log Kow is used in Table 1 and the text without explanation. It should be explained.

3.     In Figure 7, it is better to have extra experimental points for Ce between 0 and 20 mg/L for bio-C-900 to improve the fitting accuracy of isotherm equation, because the equilibrium adsorption amount increases a lot from Qe=0 for Ce=0 to Qe about 200 mg/g at Ce=20 mg/L. However, Qe only reaches about 275 mg/g at Ce=60 mg/L. More experimental points seem needed for the small concentration region.

4.     On lines 283-287, the following is mentioned:

The adsorption of SMX on bio-C-900 tends to be homogeneous monolayer adsorption [36] since the Langmuir model linear fitting result (R2 = 0.98) is slightly better than that in the Freundlich model (R2 = 0.96). On the other hand, the fitting result of bio-C-900 is better than the other two materials, indicating that bio-C-900 has a stronger possibility of heterogeneous multi-layer adsorption.

The statements are contradictory. Is the adsorption of SMX on bio-C-900 homogeneous monolayer adsorption or heterogeneous multi-layer adsorption?

5.     On lines 25-27, The removal percentage of six representative pharmaceuticals can be well correlated to the structural parameter of log Kow of each pharmaceutical, indicating the hydrophilic interaction should be one of the major mechanisms for the adsorption in water.

On line 323, Based on the above discussion, the hydrophobic properties of the materials could increase the adsorption amount.

I am confused by the above two statements.

Author Response

Response to Reviewer 2 Comments

Reviewer#2

Revisions have been made to the manuscript to accommodate reviewer 2’s comments. We have revised this manuscript considerably following the comments of Reviewer 2. Please also refer to the following part of this letter: a point-by-point response to specific comments from Reviewer 2, highlighted in yellow in the revised manuscript. We believe that this revised manuscript rectifies all the questions that Reviewer 2 raised and facilitates a better understanding. We are looking forward to your favorable decision. Your help is greatly appreciated. Please see the revised manuscript. We have revised the manuscript, which is highlighted in Yellow color.

Comment 1:

This article tries to develop an effective adsorbent for removing micropollutants in water.

The experiments seem carefully performed and well explained.

However, the following points should be clarified. 

Response 1:

We appreciate the summary of Reviewer 2. Please refer to the following part of this letter: a point-by-point response to specific comments from Reviewer 2, highlighted in yellow in the revised manuscript.

Comment 2:

The expression of Eq. (5) for Freundlich equation is incorrect.

Response 2:

We appreciate the kind suggestions of reviewer 2. We have corrected this expression and updated it in the manuscript.

Page 13, line 310 in the revised manuscript now reads:

                           (Eq. 5)

Comment 3:

log Kow is used in Table 1 and the text without explanation. It should be explained.

Response 3:

We appreciate the kind suggestions of reviewer 2. We have added an explanation of log Kow and briefly described its relation to hydrophobicity (Talanta. 97 (2012) 355-361. Analytical Chemistry 87(10) (2015) 5340-5347). We have updated the expressions in the revised manuscript.

Page 21, lines 518-524 in the revised manuscript now reads:

[44] T. Ngawhirunpat, S. Panomsuk, P. Opanasopit, T. Rojanarata, T. Hatanaka, Comparison of the percutaneous absorption of hydrophilic and lipophilic compounds in shed snake skin and human skin, Pharmazie 61(4) (2006) 331-335.

[45] H. Lin, L. Huang, Z. Gao, W. Lin, Y. Ren, Comparative analysis of the removal and transformation of 10 typical pharmaceutical and personal care products in secondary treatment of sewage: A case study of two biological treatment processes, J. Environ. Chem. Eng. 10(3) (2022). https://doi.org/10.1016/j.jece.2022.107638.

Page 4, lines 109-112 in the revised manuscript now reads:

Kow, which is the ratio of the concentration of molecules in the octanol phase to their concentration in the aqueous phase, is widely considered to indicate the hydrophobicity of the drugs [26]. A drug's hydrophobicity is indicated by a larger log Kow value [27].

Comment 4:

In Figure 7, it is better to have extra experimental points for Ce between 0 and 20 mg/L for bio-C-900 to improve the fitting accuracy of isotherm equation, because the equilibrium adsorption amount increases a lot from Qe=0 for Ce=0 to Qe about 200 mg/g at Ce=20 mg/L. However, Qe only reaches about 275 mg/g at Ce=60 mg/L. More experimental points seem needed for the small concentration region.

Response 4:

We appreciate the kind suggestions of reviewer 2. Thank you for your constructive suggestion. Figures have been updated in the revised manuscript. We have conducted experiments and supplemented the data points with their corresponding Qe at concentrations of 5 mg/L and 10 mg/L. And it was replaced by updated results in Figure 7 and Table 4. The updated R2 of Langmuir model is calculated to be 0.99. The updated KF and n of Freundlich model are calculated to be 53.23 and 2.47. Generally, the experimental data fit well with the Langmuir model. We have updated Figure 7 and Table 4 and corrected the expressions in the revised manuscript.

Figure 7 Adsorption isotherm for the removal of SMX over the three materials, including active carbon (AC), ZIF-8-C, and bio-C-900. Reaction conditions: The initial concentration of SMX = 5 mg/L ~ 80 mg/L, initial solution pH = 7, temperature = 25 °C, reaction time = 600 min.

Table 4. Langmuir and Freundlich adsorption isotherm model

Langmuir model

Freundlich model

Sample

Qmax (mg/g)

KL (L/mg)

R2

KF (mg g-1(L mg-1)1/n)

n

R2

Bio-C-900

350.90

0.061

0.99

53.23

2.47

0.96

AC

76.96

0.025

0.98

29.79

4.03

0.82

ZIF-8-C

26.44

0.027

0.99

10.88

4.32

0.88

Page 2, lines 40-46 in the revised manuscript now reads:

The saturated adsorption amount for sulfamethylthiazole (SMX) over MOF-derived carbon can reach 350.90 mg/g with a fast-initial adsorption rate of 315.29 (mg/g • min). By adding the second linker adenine as the precursor, the adsorption performance for SMX was extremely better than that of traditional active carbon (AC) and the pyrolyzed ZIF-8(ZIF-8-C), one of the most classic Zn-MOFs. The adsorption capacity calculated by the Langmuir model (R2 = 0.99) for SMX over bio-C-900 was 4.6 and 13.3 times more than that of AC and ZIF-8-C, respectively.

Page 13, lines 301-304 in the revised manuscript now reads:

At initial concentrations of 20 mg/L, 86.6% of SMX could be removed by bio-C-900 sorbent. When the initial concentrations were 40, 50, 60, and 80 mg/L, SMX was partially removed with efficiencies of 58.3 %, 53.1 %, 44.8 %, and 32.7 %, respectively.

Page 14, lines 314-327 in the revised manuscript now reads:

The adsorption of SMX on bio-C-900 tends to be homogeneous monolayer adsorption [36] since the Langmuir model linear fitting result (R2 = 0.99) is slightly better than that in the Freundlich model (R2 = 0.96). Therefore, the material tends to be more homogeneous for monolayer adsorption. In addition, the maximum adsorption capacity of SMX on bio-C-900 (350.90 mg/g) was 3.6 times higher than that on AC (76.96 mg/g), which further proved that bio-C-900 is conducive to the transport of substrates on its surface and pores. The heterogeneous multi-layer adsorption effectively improves the adsorption capacity and adsorption kinetics. The specific fitting data is shown in Table 4 below.

Figure 7 Adsorption isotherm for the removal of SMX over the three materials, including active carbon (AC), ZIF-8-C, and bio-C-900. Reaction conditions: The initial concentration of SMX = 5 mg/L ~ 80 mg/L, initial solution pH = 7, temperature = 25 °C, reaction time = 600 min.

Table 4. Langmuir and Freundlich adsorption isotherm model

Langmuir model

Freundlich model

Sample

Qmax (mg/g)

KL (L/mg)

R2

KF (mg g-1(L mg-1)1/n)

n

R2

Bio-C-900

350.90

0.061

0.99

53.23

2.47

0.96

AC

76.96

0.025

0.98

29.79

4.03

0.82

ZIF-8-C

26.44

0.027

0.99

10.88

4.32

0.88

Page 16, lines 368-369 in the revised manuscript now reads:

The adsorption capacity calculated by the Langmuir model (R2 = 0.99) for SMX over bio-C-900 was 4.6 and 13.3 times more than AC and ZIF-8-C, respectively.

Comment 5:

On lines 283-287, the following is mentioned:

The adsorption of SMX on bio-C-900 tends to be homogeneous monolayer adsorption [36] since the Langmuir model linear fitting result (R2 = 0.98) is slightly better than that in the Freundlich model (R2 = 0.96). On the other hand, the fitting result of bio-C-900 is better than the other two materials, indicating that bio-C-900 has a stronger possibility of heterogeneous multi-layer adsorption.

The statements are contradictory. Is the adsorption of SMX on bio-C-900 homogeneous monolayer adsorption or heterogeneous multi-layer adsorption?

Response 5:

Thank you for this comment. We've improved the statement in the revised manuscript. 

Page 14, lines 316-318 in the revised manuscript now reads:

Therefore, the material tends to be more homogeneous for monolayer adsorption.

Comment 6:

On lines 25-27, The removal percentage of six representative pharmaceuticals can be well correlated to the structural parameter of log Kow of each pharmaceutical, indicating the hydrophilic interaction should be one of the major mechanisms for the adsorption in water.

On line 323, Based on the above discussion, the hydrophobic properties of the materials could increase the adsorption amount.

I am confused by the above two statements.

Response 6:

Thank you for this comment. With these two studies (J. Hazard. Mater. 254-255 (2013) 345-353. Water Res. 85 (2015) 95-102.), we illustrate the respective roles assumed by the hydrophobicity of the drug and the material between the hydrophobic effect and the amount of adsorption. We have added the discussions in the revised manuscript.   

Page 21-22, lines 525-531 in the revised manuscript now reads:

[46] T.X. Bui, V.H. Pham, S.T. Le, H. Choi, Adsorption of pharmaceuticals onto trimethylsilylated mesoporous SBA-15, J. Hazard. Mater. 254-255 (2013) 345-353. https://doi.org/https://doi.org/10.1016/j.jhazmat.2013.04.003.

[47] Y. Matsui, S. Nakao, A. Sakamoto, T. Taniguchi, L. Pan, T. Matsushita, N. Shirasaki, Adsorption capacities of activated carbons for geosmin and 2-methylisoborneol vary with activated carbon particle size: Effects of adsorbent and adsorbate characteristics, Water Res. 85 (2015) 95-102. https://doi.org/https://doi.org/10.1016/j.watres.2015.08.017.

Page 2, lines 46-49 in the revised manuscript now reads:

The removal percentage of six representative pharmaceuticals can be well correlated to the structural parameter of log Kow of each pharmaceutical, indicating the hydrophobic interaction should be one of the major mechanisms for the adsorption in water.

Page 15-16, lines 350-354 in the revised manuscript now reads:

Tung XuanBui et al. performed a linear relationship between log Kow and adsorption amount at a wide range of pH for 12 drugs, indicating that the adsorption of drugs on TMS-SBA-15 is mainly driven by hydrophobic interactions [46]. Taku Matsushita [47] et al. evaluated the adsorption capacity of nine activated carbons for geosmin and 2-methylisoborneol (MIB) and showed that hydrophobic materials enhance the adsorption. Based on the above discussion, hydrophobic interactions are dominant in this adsorption mechanism.

Page 16, lines 370-371 in the revised manuscript now reads:

Hydrophobic interaction should be one of the major mechanisms for the adsorption in water. This study offers a strategy to develop novel carbon materials to remove pharmaceuticals.
